# Regenerative Superhydrophobic Coatings for Enhanced Performance and Durability of High-Voltage Electrical Insulators in Cold Climates

**DOI:** 10.3390/ma17071622

**Published:** 2024-04-02

**Authors:** Helya Khademsameni, Reza Jafari, Anahita Allahdini, Gelareh Momen

**Affiliations:** Department of Applied Sciences, University of Quebec in Chicoutimi (UQAC), 555 Boul de l’Université, Chicoutimi, QC G7H 2B1, Canada; hkhademsam@etu.uqac.ca (H.K.); aahesarouy@etu.uqac.ca (A.A.); gmomen@uqac.ca (G.M.)

**Keywords:** healing agent, superhydrophobicity regeneration, non-wettability, electrical insulators, high-voltage applications, PDMS elastomer, F-POSS

## Abstract

Superhydrophobic coatings can be a suitable solution for protecting vulnerable electrical infrastructures in regions with severe meteorological conditions. Regenerative superhydrophobicity, the ability to regain superhydrophobicity after being compromised or degraded, could address the issue of the low durability of these coatings. In this study, we fabricated a superhydrophobic coating comprising hydrophobic aerogel microparticles and polydimethylsiloxane (PDMS)-modified silica nanoparticles within a PDMS matrix containing trifluoropropyl POSS (F-POSS) and XIAMETER PMX-series silicone oil as superhydrophobicity-regenerating agents. The fabricated coating exhibited a static contact angle of 169.5° and a contact angle hysteresis of 6°. This coating was capable of regaining its superhydrophobicity after various pH immersion and plasma deterioration tests. The developed coating demonstrated ice adhesion as low as 71.2 kPa, which remained relatively unchanged even after several icing/de-icing cycles. Furthermore, the coating exhibited a higher flashover voltage than the reference samples and maintained a minimal drop in flashover voltage after consecutive testing cycles. Given this performance, this developed coating can be an ideal choice for enhancing the lifespan of electrical insulators.

## 1. Introduction

In recent years, there has been considerable attention given to the development of superhydrophobic materials. These materials, characterized by a static water contact angle greater than 150° and both a contact angle hysteresis and a sliding angle less than 10°, achieve their unique properties through the combined use of low surface energy materials and micro-nano surface roughness. Superhydrophobic materials offer significant advantages, such as water repellency and self-cleaning properties, which make them suitable for various applications, including water–oil separation, anti-corrosion, and anti-icing purposes [1,2].

A particularly interesting application of superhydrophobic coatings is the protection of electrical infrastructure, including electrical insulators. By leveraging the self-cleaning and icephobic properties of these coatings, the durability of electrical insulators can be significantly enhanced. These properties improve the insulators’ surface resistance to flashover and increase flashover voltage [3]. Applying superhydrophobic coatings to electrical insulators offers effective protection against pollution or ice accumulation. Pollution, which may include mineral matter, electronically conductive metal oxides, soluble salts, and water, can form a conductive layer on the insulators. Similarly, ice formation on the insulators, facilitated by the freezing process, corona discharge products, and the presence of other contaminating substances, can lead to increased conductivity [4]. This increase in conductivity may result in leakage currents, potentially leading to flashover formation [5].

Flashovers can lead to the loss of electrical insulation, resulting in surface damage, material degradation, and ultimately, the failure of electrical insulators. Consequently, incorporating anti-icing and self-cleaning properties in the surfaces of high-voltage insulators may become essential [6].

Therefore, researchers have concentrated on developing electrical insulators using materials with superhydrophobic properties. For example, Vazirinasab et al. [7] fabricated superhydrophobic high-temperature vulcanized (HTV) silicone rubber surfaces using atmospheric pressure plasma systems. Their goal was to enhance standard silicone-based insulators by introducing self-cleaning capabilities through the incorporation of superhydrophobicity. Similarly, Maghsoudi et al. [8] created superhydrophobic HTV silicone rubber through compression molding using a replication method. This process involved creating micro-nanostructures on the surface of HTV silicone rubber through the use of an etched aluminum surface, resulting in a superhydrophobic rubber with self-cleaning properties. These innovations represent substantial advances in producing the next generation of silicone rubber insulators.

Existing porcelain, glass, and silicone rubber insulators could be improved by applying a superhydrophobic coating. For instance, Ribeiro et al. [9] developed superhydrophobic coatings for electrical insulators using three different types of polysiloxane. They assessed the effect of these polysiloxanes on the coatings’ final properties through contact angle goniometry, adhesion tests, and resistance to tracking and erosion. Among the tested coatings, the one containing PDMS-treated nano-silica within a matrix of methyl methoxy siloxane and methyl-silsesquioxane demonstrated superior performance.

Although research has yielded important findings regarding the application of superhydrophobic coatings on electrical insulators, the durability of these coatings requires further investigation. The durability of superhydrophobic coatings is a critical factor that affects their service life and applicability. de Santos et al. [10] performed leakage current analyses and visual inspections of the surface to test the durability of their developed superhydrophobic coating. At an outdoor testing station, they determined the degradation rate of electrical insulators coated with their superhydrophobic nano-coatings through comparative leakage current analysis, surface condition monitoring, and observations of dry band arcing. They found that, under real-life conditions, the superhydrophobic coating initially performed well but began to lose its superhydrophobic properties after a short period. The nano-coated insulators initially showed a significant reduction in leakage current for the first two test quarters. However, by the end of the testing period, the coating failed to suppress leakage current, indicating a transition from hydrophobicity to hydrophilicity.

The longevity of superhydrophobic coatings poses a challenge, as surface structures featuring micro- or nano-roughness are more prone to mechanical stress than conventional surfaces. Superhydrophobicity can be compromised in two primary ways: (i) physical damage to the structure from impact or abrasion, which increases the contact area between the solid substrate and water, or (ii) an increase in surface energy because of contamination, irradiation, or damage to the hydrophobic low-surface-energy layer. In situations where use of these coatings would be ideal, they would also be expected to be relatively durable under various environmental conditions. Given the inherent fragility of existing superhydrophobic coatings, some researchers have focused on identifying design strategies to develop more robust formulations. The strategies for increasing durability are divided into passive resistance and active regeneration. Passive resistance strategies aim to preserve a surface’s superhydrophobicity after wear, possibly by reinforcing the superhydrophobic properties with techniques such as incorporating elastic compositions to absorb shock, increasing crosslinking sites, enhancing interactions between components, and improving the adhesion between the coating and substrate [11,12,13,14,15,16,17]. Active regeneration methods focus on restoring superhydrophobicity after deterioration, primarily relying on self-healing or easy repair techniques, with the former being the primary focus of this research [11].

Regenerative superhydrophobicity offers a potential solution to the durability issues associated with superhydrophobic coatings. These coatings can be defined as those capable of regaining their superhydrophobic properties through self-healing mechanisms after deterioration. The concept of self-healing is inspired by biology; for instance, lotus and clover leaves [16] exhibit self-healing superhydrophobicity by restoring their regenerable epicuticular wax layer. Regenerative superhydrophobic coatings can be classified on the basis of their healing principles and the type of damage they repair. The healing principles of damaged superhydrophobic coatings can be categorized as (i) transportation of low-surface-energy material to the surface following damage to the low-surface-energy layer because of chemical degradation; (ii) regeneration of hierarchical topography following surface damage from abrasion, scratches, etc.; and (iii) simultaneous use of both principles to repair both physical and chemical damage [18,19]. Depending on the final application or desired properties, researchers have used various low-surface-energy materials in their superhydrophobic coatings [20] (Table 1).

Among the components for regenerating superhydrophobicity, polyhedral oligomeric silsesquioxane (POSS) emerges as a suitable candidate enhancing multiple properties of a coating. POSS, a class of hybrid materials, can reduce a coating’s surface energy, increase its surface roughness, and enhance its hydrophobicity [34]. Its polyhedral structure and tunable peripheral organic groups make POSS an ideal choice for boosting a coating’s hydrophobicity, thus providing an intriguing option for self-healing superhydrophobic coatings [23,24,35]. Silicone oil, though less common in this field, serves as another agent for reducing surface energy, primarily consisting of low-molecular-weight chains with Si-O-Si bonds and surface tensions similar to PDMS [36]. Research on silicon-based coatings, such as the study by Zhu et al. [37], has demonstrated the efficacy of silicone-oil-infused polydimethylsiloxane coatings with icephobic properties. Their findings suggest that silicone oil, because of its exceptionally low surface energy, enhances the water repellency of the coating.

A comprehensive study of the fabrication of a self-healing superhydrophobic coating for electrical insulators capable of enduring real-life conditions remains a critical gap in the current research. Thus, this study aims to develop a regenerative superhydrophobic coating for electrical insulators using POSS and silicone oil as regeneration agents. The durability and regenerative nature of the coating were tested by immersing it in different pH buffer solutions and subjecting it to air plasma treatments. Its characteristics were analyzed using water contact angle, surface profilometry, X-ray photoelectron spectrometry, and FT-IR. Additional tests of flashover voltage and ice adhesion revealed promising results, indicating that the regenerative superhydrophobic coating could significantly prolong the lifespan of electrical insulators.

## 2. Materials and Methods

### 2.1. Materials and Coating Preparations

The components of the developed superhydrophobic coating are listed in Table 2, and the preparation steps are shown in Figure 1. The coating was fabricated by spin coating a silicone-based matrix, which contained nano- and microparticles along with superhydrophobicity regeneration agents, onto various substrates. Initially, 1.2 g of hydrophobic-modified silica nanoparticles were dispersed in a solvent using an ultrasonic probe. Subsequently, 0.7 g of hydrophobic silica aerogel microparticles, 0.4 g of FL0578 trifluoropropyl POSS (F-POSS) from hybrid plastics ((Hattiesburg, MS, USA), and 0.1 g of XIAMETER™ PMX-200 silicone fluid 50 cSt from Dow Corning (Midland, MI, USA) were added to the mixture and dispersed with a mechanical mixer. Sylgard 184 and its curing agent, at a 100:10 weight ratio, were then added to the mixture and stirred for an additional 5 min to form a homogeneous mixture. This mixture was then spin coated onto the substrates, and the samples were subsequently cured in a 125 °C oven for 20 min. These samples are henceforth referred to as SHP–REG. A similar formulation and preparation method, excluding the F-POSS and XIAMETER™ PMX-200 silicone fluid as superhydrophobicity regeneration agents, were used to develop superhydrophobic reference samples, coded as SHP. Reference resin samples, formulated at a 100:10 weight ratio of Sylgard 184 to the curing agent, were fabricated and are coded as SYL.

### 2.2. Characterization

The thickness of the semi-transparent coating was measured using an Elektrophysik MiniTest70 thickness gauge (ElektroPhysik, Lynbrook, NY, USA) coating thickness gauge.

The wettability of the coatings was analyzed using a DSA100 (Krüss, Hamburg, Germany) contact angle goniometer. This device measured both static and dynamic contact angles. A 4 µL water droplet was placed on the surface to measure the static contact angle, with the test repeated at least five times at different areas on the sample to obtain an average. For contact angle hysteresis measurements, a 4 μL droplet was placed on the surface and held with a needle while the substrate was moved slowly. In this setup, the contact angle at the droplet’s front is termed the advancing angle, and the receding angle is at the opposite side of the droplet [38]; their difference defines the contact angle hysteresis. The sliding angle was determined using an adjustable tilting plate, capable of tilting up to 90°. A 10 µL water droplet was placed on the sample’s surface, which was fixed on the plate. The plate was tilted until the droplet began to slide; this angle of tilt represented the sliding angle.

The chemical composition of the samples was characterized using a Cary 360 FT-IR spectrophotometer (Agilent, Santa Clara, CA, USA) in ATR (attenuated total reflection) mode.

A 3D profilometer (Profil3D Filmetrics, San Diego, CA, USA) was used to examine the surface roughness of the coatings, reporting various topographic data such as roughness average, average maximum height, maximum surface height, and valley depth.

An X-ray photoelectron spectrometer (XPS), jointly produced by Plasmionique Varennes, QC, Canada) and Staib Instruments (Langenbach, Germany) and equipped with a non-monochromatic Al anode (maximum energy 1486.6 eV), was used to analyze the surface chemical composition of the coatings.

To assess the self-healing ability of the coating, samples were exposed to air plasma until superhydrophobicity was lost. We used a Plasma Jet AS400 (Plasmatreat GmbH, Steinhagen, Germany) atmospheric-pressure plasma system with a reference voltage of 70%, a plasma voltage of 250 V, a plasma current of 16 A (power of 2.8 kW), a distance of 30 mm from the plasma nozzle to the sample surface, and a jet speed of 2 m·min^−1^. After plasma-induced deterioration, samples were placed in a 140 °C oven for 5 min before the water contact angle was measured to evaluate the regeneration of superhydrophobicity. This procedure was repeated for several cycles. We also tested the self-healing superhydrophobic coating by immersing it repeatedly in acidic (pH 2) and basic (pH 10) buffer solutions to investigate its regenerative capability. After each immersion, the water contact angle was recorded.

For flashover voltage tests, an AC voltage was applied to the samples. Before testing, coatings were spin coated on GPO3, a glass-fiber-reinforced thermoset polyester. This strong and stiff material, known for its exceptional electrical properties such as flame and arc resistance, is a common choice for electrical insulation applications. A custom-built setup was used to place the samples between two electrodes spaced 36 mm apart (Figure 2). Initially, the voltage was increased to 50% of the estimated flashover value, prior to any visible discharge activity. Subsequently, the voltage was raised at a rate of 0.5 kV·s^−1^ until the first complete discharge occurred, at which point the voltage was recorded. This test was conducted 10 times on each sample, with a 2 min pause between tests to ensure dissipation of any residual charge. We also tested the reference superhydrophobic samples and base resin reference samples using this setup. For wet conditions, deionized water was sprayed on the sample before each test.

The ice adhesion was assessed using an ice push-off test. Initially, samples were placed in a cold room at a temperature of −15 °C for 5 h. Subsequently, a thin cylindrical mold (radius 0.6 cm) was positioned on each sample and filled with deionized water. The samples remained in the cold room for an additional 24 h to ensure complete ice formation. After this period, we measured the force required to detach the cylinder from the surface [39]. The ice adhesion strength was calculated by dividing the maximum recorded force by the area of the ice cylinder in contact with the substrate.

## 3. Results & Discussion

### 3.1. Wettability

The thickness of the SHP–REG coating was approximately 100–150 µm. This semi-transparent coating exhibited a water contact angle of 169.5° and a contact angle hysteresis of 6°. A wettability comparison is presented in the Appendix A to illustrate the superhydrophobic properties of the SHP–REG coating (Appendix A). The SHP–REG coating was applied to one side of a fabric (Appendix A), whereas the other side remained untreated (Appendix A). Colored water droplets were placed on both sides of the fabric. Water droplets easily penetrated the untreated fabric, unlike the SHP–REG side, where the water-repellent layer prevented the fabric from getting wet.

The developed coating was spin coated onto filter paper, which then exhibited superhydrophobic characteristics. When the paper was submerged in colored water, it quickly floated to the surface (Appendix A).

To further highlight the self-cleaning behavior of the SHP–REG coating, we applied it to filter paper (Figure 3) and glass slides (Appendix A). The samples were dirtied with carbon black particles to simulate pollution. Using a 10 mL syringe filled with water, we demonstrated the coatings’ self-cleaning ability by adding droplets to the coated surfaces. When the droplets contacted the coating surface, they effortlessly removed the contaminants because of the water’s greater affinity for the pollutants relative to their adhesion to the superhydrophobic surface; this resulted in a clean surface (Appendix A).

### 3.2. Superhydrophobicity Regeneration

#### 3.2.1. Immersion in pH Solutions

The durability of the SHP–REG coating, as well as the SHP and SYL samples, was tested by immersing them in highly acidic and basic buffer solutions (Table 3). After 2 h of immersion in the buffer solutions, the SHP–REG coating maintained its superhydrophobicity, whereas the SHP sample did not. Nonetheless, after 12 h of immersion, the SHP–REG sample also lost its superhydrophobicity, displaying a water contact angle of less than 150°. Remarkably, the SHP–REG coating regained its superhydrophobicity after 24 h at ambient temperatures, a recovery not observed for the SHP sample. This loss and subsequent regain of superhydrophobicity is attributed primarily to an increase in the surface energy of the sample, which, after being kept at ambient temperature, allowed the superhydrophobic regeneration agents to migrate to the surface, thus reducing the surface energy and restoring superhydrophobicity. A slight increase in the water contact angle observed in the SHP and SYL samples may reflect the hydrophobic recovery property of PDMS-based coatings [40].

#### 3.2.2. Plasma Treatment

To confirm the superhydrophobicity regeneration capabilities of the SHP–REG samples, they were subjected to air plasma treatment for 8 min [41]. Figure 4 illustrates the significant reduction in water contact angle on the deteriorated coating following plasma exposure. This exposure introduces hydrophilic oxygen-containing groups to the surface [22,42], transforming it from superhydrophobic to hydrophilic and indicating a chemical modification because of interactions with oxygen and nitrogen radicals and ions as well as the introduction of polar groups [43]. However, after heating to 140 °C for 5 min or being left at an ambient temperature for 12 h, the SHP–REG samples regained their superhydrophobicity.

The durability of the SHP–REG coatings’ self-healing properties was further tested by subjecting them to several cycles of plasma deterioration and healing (Figure 5). Both the SHP–REG and SHP coatings underwent at least six plasma treatment cycles. After each cycle, the contact angle for both samples decreased. Following the initial plasma exposure, the SHP samples failed to recover their superhydrophobicity, with a contact angle recovery to only 120°. Conversely, the SHP–REG samples successfully regained a contact angle above 150° after each plasma exposure, maintaining superhydrophobicity even after nearly six cycles, with a contact angle of 154° documented (Appendix A). This regenerative ability of the SHP–REG coatings occurs because of the migration of silicone oil and F-POSS to the surface, leading to a reduced surface energy [24,37].

### 3.3. Surface Characteristics

Surface morphology significantly influences the development of superhydrophobic coatings [38,44]. Thus, to evaluate the impact of plasma exposure on the SHP–REG surface, profilometry analyses were conducted (a) before any damage, (b) after plasma exposure resulting in superhydrophobicity loss, and c) following superhydrophobicity regeneration after 12 h at room temperature (Figure 6 and Table 4).

The root mean square (RMS) of roughness represents the standard deviation of the surface roughness height distribution. Before plasma exposure, the SHP–REG coating exhibited an excellent micro-nanostructure with significant RMS and arithmetic average (Sa) roughness values of 306.8 nm and 175 nm, respectively. This hierarchical structure was largely preserved post-plasma exposure, with RMS and Sa values reduced slightly to 225 nm and 150.3 nm, respectively. The hierarchical micro-nanostructures facilitated the trapping of air beneath the water droplets, leading to the creation of superhydrophobic surfaces. A comparison of the 3D profiles before and after plasma treatment reveals that the loss of hierarchical roughness was minimal. As hierarchical roughness is essential for maintaining superhydrophobicity, this minimal deterioration allowed for the regeneration of superhydrophobicity for the SHP–REG samples (Appendix A).

FT-IR spectra of the SHP–REG coating were produced before any damage, after plasma deterioration leading to a loss of superhydrophobicity, and after superhydrophobicity regeneration (Figure 7). Peaks at 1259 cm^−1^ and 795 cm^−1^ represent Si-CH_3_ and Si-C bonds in the coating, respectively, and the peak around 2950 cm^−1^ was because of asymmetric CH_3_ stretching in Si-CH_3_ [45,46]. The CH_3_ peak was evident in the developed sample before any plasma-related deterioration. However, the plasma-degraded sample showed a significant reduction in the intensity of this function (Figure 7). Peak intensity increased in the samples with regenerated superhydrophobicity, suggesting that the surface chemistry of the samples is susceptible to changes during plasma-induced deterioration.

Comparing the spectra of the pristine superhydrophobic sample and the regenerated superhydrophobic sample reveals no significant difference. The negligible difference between the sample before plasma treatment and after superhydrophobicity regeneration indicates that the sample regained its original surface chemistry. Nevertheless, a more precise investigation is required to understand the changes that the developed coating underwent. Therefore, the samples were also analyzed by XPS.

To understand the self-healing mechanism governing the superhydrophobicity of the SHP–REG sample, we performed XPS analysis. All binding energy values were calibrated using the reference peak of C1s at 284.8 eV. The XPS spectra of the samples (Figure 8) demonstrate that the SHP–REG sample withstood an increase in the O 1s signal after plasma-related deterioration. The plasma treatment likely generated hydroxyl groups on the coating surface, causing changes in O 1s signals [47]. The signal decreased in the SHP–REG coating after superhydrophobicity regeneration, showing that the rearrangement of the coating components covered the hydroxyl groups and that other carbon–oxygen polar groups could have formed [48].

The curve fittings of C1s for SHP–REG before plasma (Figure 9), after plasma deterioration, and after superhydrophobicity regeneration consisted of four peaks: the peak at 274 eV corresponding to C-Si, the peak at 284.8 eV corresponding to C–C and C–H moieties, and the peak at 286.4 eV corresponding to C–OH and C–O–C functional groups [43,49,50]. The ratio of C–OH and C–O–C in the SHP–REG sample before plasma deterioration was approximately 6.1%, which increased to 10.8% after plasma deterioration. However, this ratio dropped to around 7.8% for the sample after superhydrophobicity regeneration. Therefore, the increase in hydrophilic polar oxygen-containing groups on the surface after air plasma justifies the decrease in the water contact angle of these samples.

### 3.4. Flashover Voltage

To assess the efficiency of the developed SHP–REG coating intended for electrical insulators, we tested the dielectric strength of the samples. A flashover voltage test was performed to assess the performance of the SHP–REG, SHP, and SYL samples. In the dry test, the SHP–REG coating maintained its flashover voltage over 10 cycles (Figure 10a). The flashover voltage under wet conditions was highest for the SHP–REG sample (Figure 10b). The water repellency of the surface prevented the formation of a thin water film on the superhydrophobic coatings, allowing separate water droplets to be clearly visible on the surface. Therefore, the SHP–REG coating exhibited nearly identical electrical properties to those observed in the dry test. When the flashover test was repeated, the superhydrophobicity of the SHP–REG and SHP coatings was compromised, resulting in less dispersed water droplets on the surface, thereby forming a water film. Hence, we observed an increased flashover voltage, and by the end of the 10th test, the flashover voltage of both SHP–REG and SHP samples resembled that of the SYL sample.

Both superhydrophobic coatings, SHP–REG and SHP, demonstrated good performance in the flashover test under wet conditions. Thus, the water repellency of the superhydrophobic coatings helped reduce or delay flashover-related issues. The flashover voltage of the SHP sample significantly decreased after three tests, whereas the regenerative superhydrophobic coating (SHP–REG) experienced a notable drop in flashover voltage only after the fifth test. The presence of superhydrophobic regenerative agents in this coating effectively delays the effect of flashover on the coating’s nonwettability. If the low-surface-energy layer of the coating is affected during the test, the superhydrophobicity regeneration agents can migrate to the surface and delay the formation of a water film on the coating.

### 3.5. Ice Adhesion

The ice adhesion strength of the SHP–REG, SHP, and SYL samples was determined using a push-off test (Table 5). Compared to the SHP coating, the SHP–REG coating demonstrated a significantly lower ice adhesion strength, which can be attributed to the presence of silicone oil, which increases the surface’s slipperiness [51].

Ice adhesion was then evaluated over multiple icing/de-icing cycles (Figure 11). For the SHP–REG coating, an increase in ice adhesion after each icing/de-icing cycle was expected given the depletion of silicone oil on the surface [51]. However, contrary to expectations, the SHP–REG coating exhibited a lower ice adhesion after each cycle. This phenomenon can be explained by the depletion of silicone oil on the surface in each cycle, which leads to an increase in the surface energy of the coating. This triggers the migration of F-POSS and silicone oil to the surface to reduce the surface energy, resulting in a further decrease in the ice adhesion strength of the coating.

## 4. Conclusions

The presence of ice or pollution on electrical infrastructure, such as electrical insulators, can disrupt their functionality. Superhydrophobic coatings could be an appropriate solution to this issue; however, their low durability is a significant drawback. This research addresses a gap in existing literature within the domain. The development of a regenerative superhydrophobic coating, designed for application on electrical insulators and infused with select superhydrophobicity agents, capitalizes on their combined presence. The fabrication of regenerative superhydrophobic coatings capable of regaining superhydrophobicity after damage to the superhydrophobic layer would be an effective response to this challenge. We therefore developed a PDMS-based superhydrophobic coating containing hydrophobic aerogel microparticles and fumed silica and post-treated with polydimethylsiloxane nanoparticles. It is essential to incorporate self-healing superhydrophobic agents into the coating to achieve superhydrophobicity regeneration. Because of the significant potential of F-POSS in enhancing the coating’s hydrophobicity and silicone oil in reducing surface energy, FL0578 trifluoropropyl POSS (F-POSS) and XIAMETER PMX-series silicone oil were added to the coating mixture. The coating was applied to substrates using spin coating. The final coating, coded as SHP–REG, demonstrated a contact angle as high as 169.5 ± 0.6° and exhibited excellent water repellency and self-cleaning properties after water immersion and dry contamination tests.

The superhydrophobicity regeneration of the SHP–REG sample was verified through repeated immersion in acidic and basic buffer solutions and repeated air plasma treatments. In contrast to the SHP coating, the developed coating was able to regain superhydrophobicity after both tests, indicating a favorable performance for its overall application. After each plasma treatment, the SHP–REG coating recovered its superhydrophobicity autonomously. The superhydrophobicity regeneration capability, determined by water contact angle measurements, profilometry, and FT-IR analysis, confirmed the regeneration of the coating’s superhydrophobicity.

To better evaluate the performance of the developed coating for real-life application on electrical insulators, we ran a series of experiments. The flashover test demonstrated the beneficial impact of regenerative superhydrophobicity in increasing or maintaining flashover voltage, especially under wet conditions.

The ice adhesion strength of the developed coatings was investigated through two approaches using the ice push-off test. Over multiple icing/de-icing cycles, the SHP–REG coating maintained a relatively low ice adhesion.

## Figures and Tables

**Figure 1 materials-17-01622-f001:**
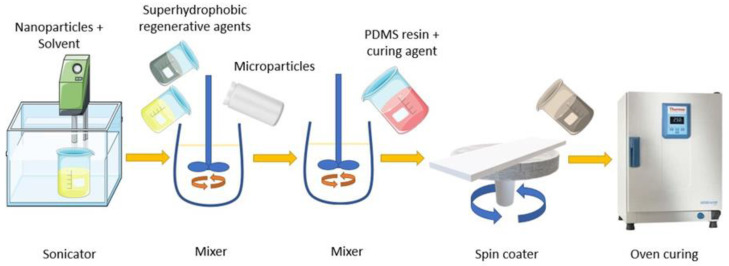
Schematic preparation steps of the SHP–REG samples.

**Figure 2 materials-17-01622-f002:**
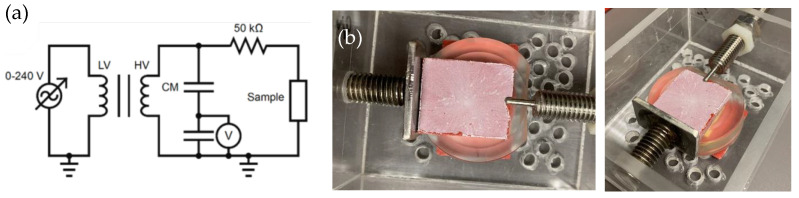
(**a**) Electrical circuit for the flashover voltage measurements [6]; (**b**) top and side views of the flashover setup preparation for the SHP–REG samples.

**Figure 3 materials-17-01622-f003:**
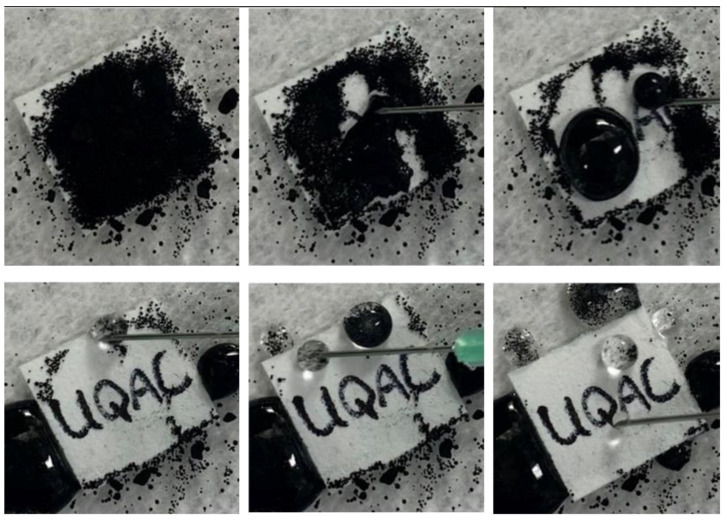
Self-cleaning evaluation of the SHP–REG coating showing the carbon black particles adhering to the added droplets.

**Figure 4 materials-17-01622-f004:**
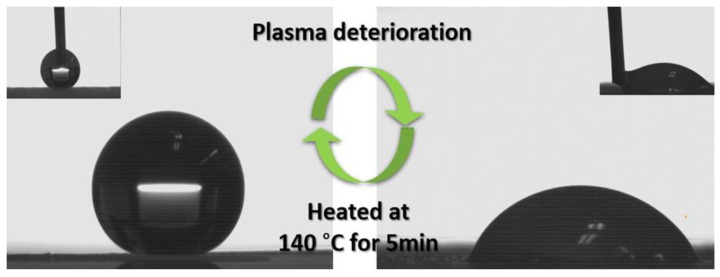
Loss of superhydrophobicity after plasma deterioration followed by superhydrophobicity regeneration for the SHP–REG samples.

**Figure 5 materials-17-01622-f005:**
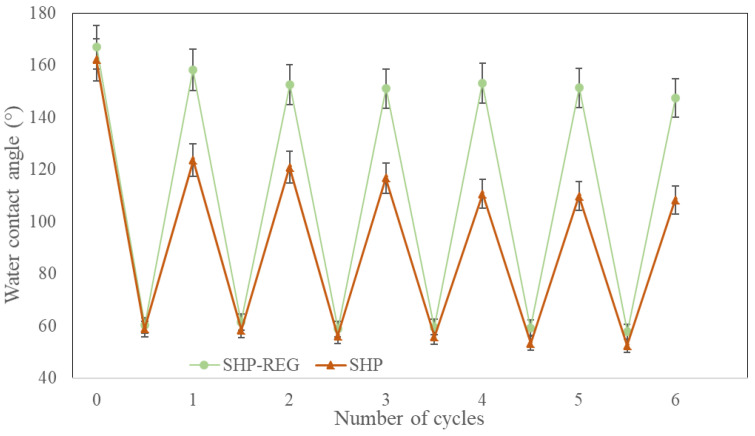
Wettability of the SHP–REG and SHP samples after air plasma deterioration and the subsequent self-healing process over multiple cycles.

**Figure 6 materials-17-01622-f006:**
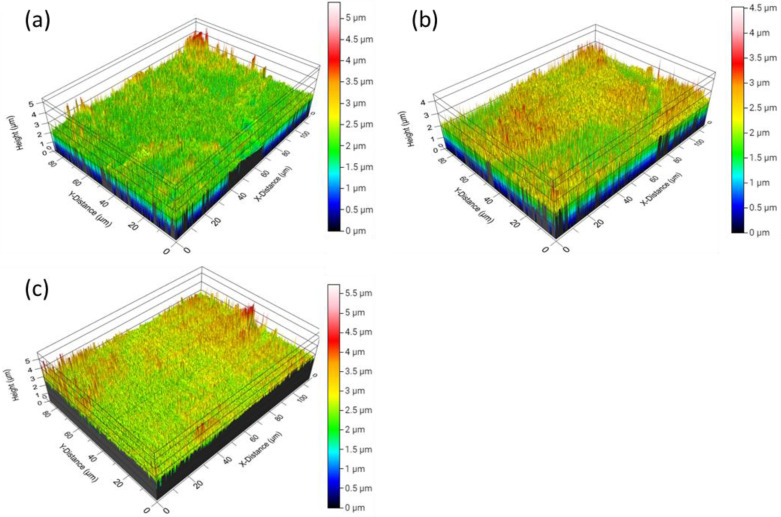
3D surface profiles in tilted view of the SHP–REG sample (**a**) prior to any damage; (**b**) after plasma deterioration, leading to the loss of superhydrophobicity; and (**c**) after the regeneration of superhydrophobicity.

**Figure 7 materials-17-01622-f007:**
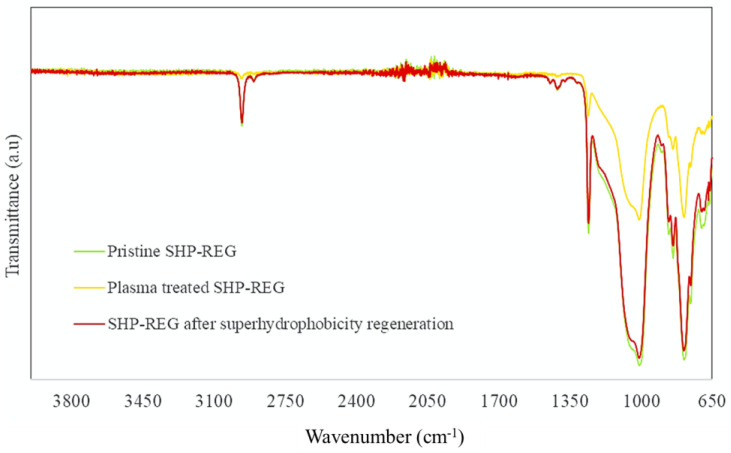
FT-IR spectra of the SHP–REG sample prior to any damage (green), after plasma deterioration leading to a loss of superhydrophobicity (yellow), and after superhydrophobicity regeneration (red).

**Figure 8 materials-17-01622-f008:**
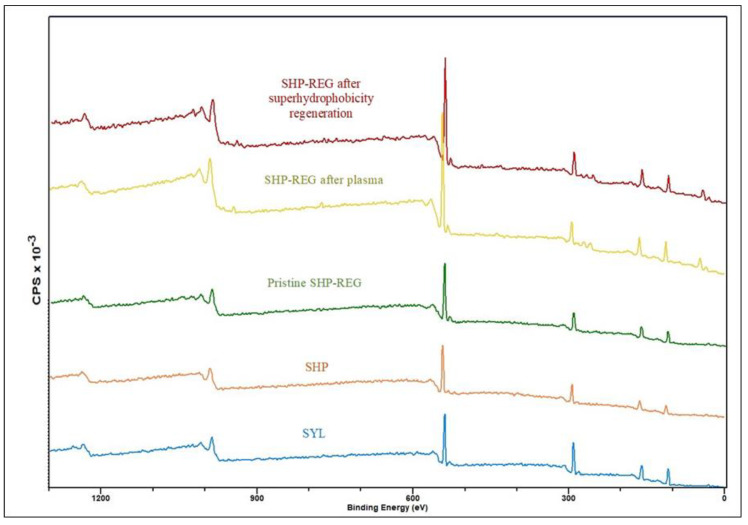
Survey XPS spectra of SHP and SYL samples and the SHP–REG coating prior to and after plasma deterioration and after superhydrophobicity regeneration.

**Figure 9 materials-17-01622-f009:**
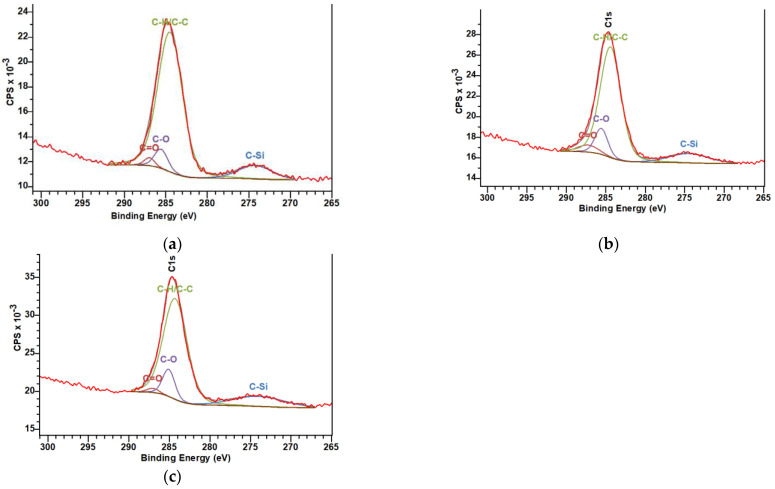
C1s spectra of SHP–REG (**a**) prior to any damage, (**b**) after plasma deterioration, and (**c**) after superhydrophobicity regeneration.

**Figure 10 materials-17-01622-f010:**
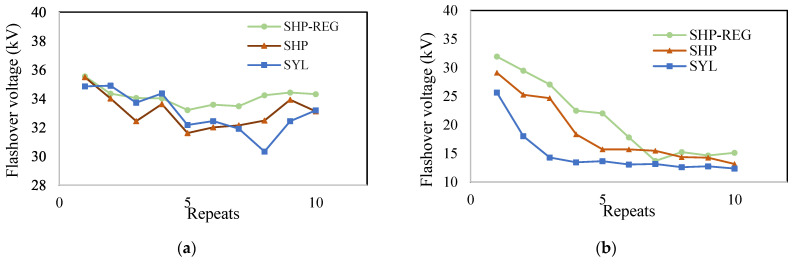
Ten repeated flashover tests in (**a**) dry and (**b**) wet conditions for the SHP–REG, SHP, and SYL samples.

**Figure 11 materials-17-01622-f011:**
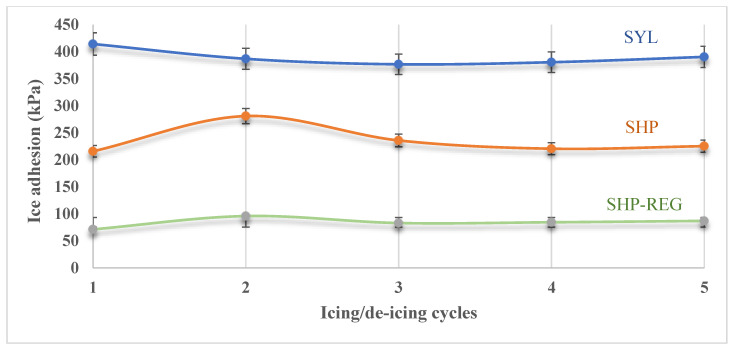
The ice adhesion strength of the SHP–REG, SHP, and SYL samples over five icing/de-icing cycles.

**Table 1 materials-17-01622-t001:** Commonly used superhydrophobic regenerative components.

Healing Agent	Chemical Compound	References
PFOS/POTS	Perfluorooctanesulfonic/Perfluorooctyltriethoxysilane	[21,22]
POSS	Polyhedral oligomeric silsesquioxane	[23,24]
FAS	Fluorinated alkyl silane	[25,26]
POTS	Perfluorooctyltriethoxysilane	[27]
POS	Polysiloxane	[28]
Beeswax	–	[29]
HDI	Hexamethylene diisocyanate (HDI)	[12]
n-Nonadecane	–	[30]
n-Octadecane	–	[31]
ODA	Octadecylamine	[32,33]

**Table 2 materials-17-01622-t002:** Materials used in this research and their function.

Material Name	Function	Company
SYLGARD™184 silicone elastomer	Matrix	Dow Corning
AEROSIL^®^ R 202	Nanoparticles	Evonik Co.
Aerogel Enova IC3 100	Microparticles	Cabot Aerogel
Hexane	Solvent	Fisher Scientific
XIAMETER™ PMX-200 silicone fluid 50	Self-healing agent	Dow Corning
FL0578 Trifluoropropyl POSS	Self-healing agent	Hybrid plastics
Glass, aluminum, fabric, paper, GPO3	Substrates	–

**Table 3 materials-17-01622-t003:** Effect of immersion in various pH solutions on the developed coatings.

Sample	pH	Initial WCA (°)	WCA (°) after 2 h Immersion	WCA (°) after 12 h Immersion	WCA (°) after 24 h at Room Temperature
SYL	10	110.3 ± 1.5	105.7 ± 0.8	97.8 ± 0.7	104.5 ± 0.4
	2	110.3 ± 1.5	108.2 ± 0.6	100.3 ± 1.9	107.3 ± 0.7
SHP	10	162.2 ± 0.8	147 ± 1.4	140.1 ± 2.1	147.9 ± 0.6
	2	162.2 ± 0.8	151 ± 1.2	142.8 ± 0.8	148 ± 1.4
SHP–REG	10	169.5 ± 0.6	159 ± 0.6	143 ± 1.7	165.9 ± 0.6
	2	169.5 ± 0.6	163 ± 0.4	148 ± 0.4	166.4 ± 1.4

**Table 4 materials-17-01622-t004:** Roughness values of the pristine superhydrophobic and plasma-treated coating.

Roughness Parameter (nm)	SHP–REG	Plasma-Deteriorated SHP–REG	Plasma-Deteriorated SHP–REG after 12 h at Room Temperature
RMS (S_Q_)	306.8	225	272.2
Mean roughness (S_A_)	175	150.3	155.9
Maximum peak height (S_P_)	3110	2241	2532
Maximum pit depth (S_V_)	2270	2290	2285

**Table 5 materials-17-01622-t005:** Ice adhesion strength of the SHP–REG, SHP, and SYL samples.

Sample	Ice Adhesion (kPa)
SHP–REG	71.2
SHP	215.6
SYL	414.1

## Data Availability

Data are contained within the article.

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
