# Peer review of "Regenerative Superhydrophobic Coatings for Enhanced Performance and Durability of High-Voltage Electrical Insulators in Cold Climates"

_materials, 2024, doi:10.3390/ma17071622_

Round 1

Reviewer 1 Report

Comments and Suggestions for Authors

This manuscript aimed to investigate the development of regenerative superhydrophobic coatings for electrical insulators, addressing the challenge of their durability and functional integrity in the presence of ice or pollution. Utilizing a PDMS-based formula as the matrix, enriched with hydrophobic aerogel microparticles, fumed silica as nanoparticles, adding extra hydrophobic self-healing agents F-POSS and silicone oil, the study demonstrates the coating's ability to restore its superhydrophobic properties after damage. Applied via spin coating, the resultant SHP-REG coating exhibited exceptional water repellency, self-cleaning abilities, and a contact angle of 169.5±0.6°. The coating's performance in regaining superhydrophobicity after exposure to various pH levels and plasma deterioration was highlighted, alongside its potential to enhance flashover voltage in electrical insulators under wet conditions and maintain low ice adhesion through multiple icing/de-icing cycles, showcasing a promising solution for enhancing the resilience and efficiency of electrical infrastructure.

The manuscript is well-structured and strategically laid out, focusing keenly on the fabrication and characterization phases. The authors employ thorough physical and chemical characterization techniques to systematically investigate the dynamics of superhydrophobicity loss and recovery under stringent conditions and after repeated tests.

Several inquiries arise from the paper that merit the authors' attention in their revision. Clarifying these points will greatly enhance the manuscript's accessibility and scientific rigor. 

• Line 42-43: It would be beneficial to provide a detailed explanation of how anti-icing or enhanced hydrophobicity contributes to improved resistance against flashover, especially for readers/audience unfamiliar with this field.

• Line 72: Could you clarify what is meant by "pollution performance"?

• Line 79: What factors contribute to the observed negative transformation in your work, and are there any insights or implications derived from this?

• Line 202: To assess regeneration, or rather anti-deterioration, is it necessary to measure the water contact angle immediately after plasma treatment, without any further treatment? In real-world scenarios, it is impractical to subject materials to conditions of 140°C for regeneration.

• Line 237: How many specimens were tested to obtain those two values? Are there average and standard deviation values available for WCA and contact angle hysteresis? Additionally, are there WCA testing images available to illustrate the calculated values, similar to Figure 6?

• Line 281-282: What happens when REG agents deplete after frequently migrating to the surface?

• Figure 7: This figure suggests that the composite SHP can achieve self-healing with just nano and micro particles, even without REG, although not achieving superhydrophobicity. Is it possible that the self-healing capability originates from these particles, independently of REG?

• Line 354-357: There is no Fig 9a, b, and c. Highlighting the color difference should suffice. Please correct this typo.

• Figure 12: Figure 12b suggests that repeated flashover voltage tests can somehow damage the superhydrophobicity of the SHP-REG coating, particularly after 5 cycles. Has your team examined the contact angle and surface morphology afterwards? It would be useful to explain why this capability decreases significantly after 1-5 cycles of testing.

Addressing these questions and suggestions will undoubtedly enrich the manuscript, emphasizing its contribution to the field. The research already stands as a valuable manuscript offering essential insights and data. With these enhancements, it promises to be a significant scientific work. Good luck with your revision.

Comments on the Quality of English Language

The quality of English writing in this work is sufficient to publish

Author Response

First of all, on behalf of my co-authors and myself, I would like to kindly thank you for the time you dedicated to read our work and make some insightful comments and advice. The comments were precise, detailed, and they aided us in improving the manuscript. We did our best to address the issues raised by the reviewers and provide an answer to all their concerns.

Our corrections and modifications are highlighted in yellow within the text.

  • Line 42-43: It would be beneficial to provide a detailed explanation of how anti-icing or enhanced hydrophobicity contributes to improved resistance against flashover, especially for readers/audience unfamiliar with this field.

R: Thank you for this comment. Indeed a clear explanation could help with better explanation of importance of these superhydrophobic coatings. The added part is highlighted in yellow in the revised manuscript.

  • Line 72: Could you clarify what is meant by "pollution performance"?

R: Pollution performance analysis would refer to an investigation of the behavior of the insulators in natural operation environment. Usually in these tests, the installed insulators are continuously tested and undergo the same operation conditions as the insulators in the line. During the period of test, different parameters such as surface leakage current, surface contaminants, meteorological conditions and visual changes are monitored [1,2]. The manuscript is modified now to demonstrate better clarification.

[1]        T.C. Cheng, C.T. Wu, J.N. Rippey, F.M. Zedan, POLLUTION PERFORMANCE OF DC INSULATORS UNDER OPERATING CONDITIONS, 1981.

[2]        H. de Santos, M. Sanz-Bobi, Research on the pollution performance and degradation of superhydrophobic nano-coatings for toughened glass insulators, Electric Power Systems Research 191 (2021) 106863. https://doi.org/10.1016/j.epsr.2020.106863.

  • Line 79: What factors contribute to the observed negative transformation in your work, and are there any insights or implications derived from this?

R: Remarkable question. As research plan, we had structured for this research, the goal was to develop a coating that was able to demonstrate superhydrophobicity regeneration with an aimed application on insulators. Therefor the mentioned coating has to show remarkable performance under tests like flashover voltage test. The next step for this research would be to take the existing developed coating and try to investigate the behavior during the flashover test and try to better comprehend the root causes and by coating optimisation have an attempt in overcoming them.

The factors contributing to the negative transformation would to our knowledge, depend on the superhydrophobicity healing principals. With further investigation we could focus on the effect of the flashover test on the micro nano roughness and low surface energy layer, by doing so we would be able to pin out the culprit and therefor be able to optimise the coating for better performance.

  • Line 202: To assess regeneration, or rather anti-deterioration, is it necessary to measure the water contact angle immediately after plasma treatment, without any further treatment? In real-world scenarios, it is impractical to subject materials to conditions of 140°C for regeneration.

R: Figure 7. illustrates a graph of two samples, SHP-REG and SHP, going under air plasma deterioration. In the graph the water contact angle measurement of the samples right after plasma deterioration are shown. The goal is to compare the performance of the two samples in regaining the water contact angle they had possessed before the plasma deterioration.

Indeed the high temperatures could not be practical for real-world scenarios, however use of this method could give us a good grasp of the superhydrophobicity regeneration abilities of the developed coating. However, as you have mentioned, there is need to be able to understand how practical the developed coating could be in real-world applications. Therefor, during the tests, we put plasma-deterioriated samples in ambient temperature and measured the water contact angle on the plasma deteriorated sections in different time frames to be able to report the superhydrophobicity regeneration abilities of the developed coating in ambient temperature. (line 301)

  • Line 237: How many specimens were tested to obtain those two values? Are there average and standard deviation values available for WCA and contact angle hysteresis? Additionally, are there WCA testing images available to illustrate the calculated values, similar to Figure 6?

R: Each contact angle reported had been tested on 3 different samples and for each sample, 5 different points were measured.

 Line 281-282: What happens when REG agents deplete after frequently migrating to the surface?

R: The amount of superhydrophobicity regeneration agents have been optimised during the coating fabrication. Depending on the method that ends in loss of superhydrophobicity, the superhydrophobicity regeneration nature of the developed coating tries to regenerate this aspect. In this example, immersion in different pH solutions, targets the low surface energy layer. After being taken out of the solution, the embedded low-surface energy material in the coating migrates to the surface. However, as mentioned in your question, it could insightful for the next step of this research to take an step even further, aiming to test the limits of superhydrophobicity regeneration by repeating the test until the coating will  not be able to regenerate its superhydrophobicity.

  • Figure 7: This figure suggests that the composite SHP can achieve self-healing with just nano and micro particles, even without REG, although not achieving superhydrophobicity. Is it possible that the self-healing capability originates from these particles, independently of REG?

R: The nature of the silicone-based coating are the long chain molecules which have a certain freedom to migrate within the developed coating. The changes shown in figure 7 for the SHP coating, refers to the nature of this silicone-based coating. However as you can see, since the low surface energy layer is demolished, these changes will not be sufficient to bring back the superhydrophobicity. Also, as you can see, after each cycle, the ability of the SHP coating to be lessens each cycle after the other. So in general, the self-healing ability can be translated in free movement of particles toward the surface. The tricky part and at the same time the artistic side of this research would be to finding components that their movement toward the surface of the coating could end up in superhydrophobicity regeneration, alongside other properties that they would bring upon the developed coating.

  • Line 354-357: There is no Fig 9a, b, and c. Highlighting the color difference should suffice. Please correct this typo.

R:  Thank you for this comment. The changes are highlighted in yellow.

  • Figure 12: Figure 12b suggests that repeated flashover voltage tests can somehow damage the superhydrophobicity of the SHP-REG coating, particularly after 5 cycles. Has your team examined the contact angle and surface morphology afterwards? It would be useful to explain why this capability decreases significantly after 1-5 cycles of testing.

R:  Thank you for this comment. This analysis and more coating optimisation based on the tests presented in this work will be next step of this research. It should be noted that the test was repeated 10 times on each sample, with a 2-minute pause between each test. However, it appears that a 2-minute pause may not be sufficient for the migration of healing agents on the surface to facilitate regenerative surface properties.

Reviewer 2 Report

Comments and Suggestions for Authors

The work presented by Helya Khademsameni et al and entitled: "Regenerative Superhydrophobic Coatings for Enhanced Performance and Durability of High-Voltage Electrical Insulators in Cold Climate" is of interest to the readers of materials, MDPI. The authors have conducted a lot of work to prepare their superhydrophobic regenerative coatings and test their applicability as electrical insulators in cold climates. Although this work is very interesting and has potential applications, the authors need to rearrange their article in a more organised way. The article seems at the current state a compilation of data or a lengthy report. I believe this paper can be accepted for publication in Materials after Major revisions. The authors are advised to consider the comments addressed below: 

(1) Abstract: It is abit long and could be slightly shortened. Also sentence: ". With its demonstrated performance, the developed coating could be an ideal choice for enhancing the lifespan of electrical insulators. You cannot start a sentence using "with". 

(2) Introduction (Line 65): "compared to the other coatings": cite the recent article: 

https://doi.org/10.1021/acsapm.3c01740

(3) In the materials and methods section, the authors need to mention the chemical supplier.

(4) Figure 3. Where is the a) and b) in the figure ?

(5) Figure 4 is repetitive to Fig 3. What does it add more compared to Fig 3 ? Additionally each sub figure needs to be labeled. 

(6) Fig 11 (XPS): The bonds need to be adjusted far abit from the peak and should have the same color of the XPS peak fitting. How was the fitting done ?

(7) Fig. 9 & 12 (poor resolution and quality). Need enhancement. avoid using grey axes. 

(8) The authors include to many figures and I believe many figures can go to the supporting information section. 

(9) Why didn't the authors also perform SEM-EDX of the coating ?

(10) What about the cost of those coatings ? Can they get commercialized in the coatings industry. The authors need to discuss this in the introduction section. 

Comments on the Quality of English Language

Major revision: The work is indeed novel and interesting but the results and discussion need to be discussed in a more organized way. Additionally some figures need to be moved to the supporting information. 

Author Response

First of all, on behalf of my co-authors and myself, I would like to kindly thank you for the time you dedicated to read our work and make some insightful comments and advice. The comments were precise, detailed, and they aided us in improving the manuscript. We did our best to address the issues raised by the reviewers and provide an answer to all their concerns.

Our corrections and modifications are highlighted in yellow within the text.

(1) Abstract: It is abit long and could be slightly shortened. Also sentence: ". With its demonstrated performance, the developed coating could be an ideal choice for enhancing the lifespan of electrical insulators. You cannot start a sentence using "with". 

R: Thank you for your comment. The changes are highlighted in the manuscript.

(2) Introduction (Line 65): "compared to the other coatings": cite the recent article: 

https://doi.org/10.1021/acsapm.3c01740

R: This article was added as ref to revised manuscript.

(3) In the materials and methods section, the authors need to mention the chemical supplier.

R: All the chemical suppliers are indicated in Table 2.

(4) Figure 3. Where is the a) and b) in the figure ?

R: Thank you for this point. The picture is modified in the revised manuscript.

(5) Figure 4 is repetitive to Fig 3. What does it add more compared to Fig 3 ? Additionally each sub figure needs to be labeled. 

R: Figure 4 shows the water-repellency of the developed coating and figure 3 shows the self-cleaning nature of this superhydrophobic coating, both valuable properties which have their specific applications in various fields. We moved this figure to supporting information section.

(6) Fig 11 (XPS): The bonds need to be adjusted far abit from the peak and should have the same color of the XPS peak fitting. How was the fitting done ?

R: the requested modifications were done in revised manuscript (figure 9 in revised manuscript)

(7) Fig. 9 & 12 (poor resolution and quality). Need enhancement. avoid using grey axes. 

R: The pictures are enhanced and modified in the revised version.

(8) The authors include to many figures and I believe many figures can go to the supporting information section. 

R: Thank you for your comment. We move figure 3 and 4 to the supporting information section.

(9) Why didn't the authors also perform SEM-EDX of the coating ?

R: Since we conducted several chemical characterizations such as FTIR and XPS, which are more accurate than SEM-EDX, we opted not to perform this particular test.

(10) What about the cost of those coatings ? Can they get commercialized in the coatings industry. The authors need to discuss this in the introduction section. 

R: Thank you for your comment. Since this study was conducted at the academic research level and the results were obtained on a laboratory scale, we are unable to discuss the cost of the product at this time. 

Comments on the Quality of English Language

Major revision: The work is indeed novel and interesting but the results and discussion need to be discussed in a more organized way. Additionally some figures need to be moved to the supporting information. 

R: Thank you for your comment. The requested modifications have been implemented in the revised manuscript.

Reviewer 3 Report

Comments and Suggestions for Authors

While this paper exhibits interest, it is observed to lack novelty, and the absence of control studies is notable. A more in-depth discussion on the uniqueness of this work is required. Additionally, addressing the following crucial details could improve the paper's potential for reconsideration for publication. 

1.      Discuss more about the novelty of this work in the introduction or conclusion part.

2.      How does the morphology of this material change after plasma deterioration and regeneration? More systematic study is required, such as FESEM analysis of bare and coated fabric. The authors are suggested to review the following articles for inspiration and cite them: a)  Robust and self-healable bulk-superhydrophobic polymeric coating, Chemistry of Materials 29 (20), 8720-8728, b) Customizing oil-wettability in air—without affecting extreme water repellency, Nanoscale 12 (48), 24349-24356.

3.      What is the contact angle of the bare fabric? Is it the same as the plasma-deteriorated one?

4.      Is Figure 8 an AFM analysis? Why does the surface roughness decrease after the regeneration of superhydrophobicity? Please provide the Root Mean Square (RMS) roughness for better understanding.

5.      How durable is this material? Authors are suggested to perform several physical and chemical durability test.

6.      Authors are suggested to include sliding angle changes along with contact angle in Figure 7 to comprehend the repeatably regenerative non-adhesive property.

Comments on the Quality of English Language

 Minor editing of English language required

Author Response

First of all, on behalf of my co-authors and myself, I would like to kindly thank you for the time you dedicated to read our work and make some insightful comments and advice. The comments were precise, detailed, and they aided us in improving the manuscript. We did our best to address the issues raised by the reviewers and provide an answer to all their concerns.

Our corrections and modifications are highlighted in yellow within the text.

While this paper exhibits interest, it is observed to lack novelty, and the absence of control studies is notable. A more in-depth discussion on the uniqueness of this work is required. Additionally, addressing the following crucial details could improve the paper's potential for reconsideration for publication. 

1-Discuss more about the novelty of this work in the introduction or conclusion part.

R: Thank you for your comment. The novelty of this research has been incorporated and highlighted in the conclusion section of the revised manuscript.

 2- How does the morphology of this material change after plasma deterioration and regeneration? More systematic study is required, such as FESEM analysis of bare and coated fabric. The authors are suggested to review the following articles for inspiration and cite them: a)  Robust and self-healable bulk-superhydrophobic polymeric coating, Chemistry of Materials 29 (20), 8720-8728, b) Customizing oil-wettability in air—without affecting extreme water repellency, Nanoscale 12 (48), 24349-24356.

R: We included an image of fabric as an application example to demonstrate that our coating is not limited to flat surfaces such as aluminum or glass. Since the fabric was not included in our samples for other characterizations, further tests were not conducted on it. Additionally, we have incorporated the suggested articles (references ...) into our revised manuscript.

3-What is the contact angle of the bare fabric? Is it the same as the plasma-deteriorated one?

R: The water contact angle of the fabric remained below 10° both before and after the plasma treatment.

  1. Is Figure 8 an AFM analysis? Why does the surface roughness decrease after the regeneration of superhydrophobicity? Please provide the Root Mean Square (RMS) roughness for better understanding.

R: These images were acquired through profilometry analysis. The root mean square (RMS) values have been included in the manuscript.

  1. How durable is this material? Authors are suggested to perform several physical and chemical durability test.

R:  We investigated the most severe conditions to demonstrate the stability of these coatings, including resistance to plasma deterioration, exposure to various pH solutions, multiple icing/de-icing cycles, and flashover tests. Additionally, we conducted several chemical characterizations, including XPS, FTIR, profilometry, contact angle measurements, and others.

  1. Authors are suggested to include sliding angle changes along with contact angle in Figure 7 to comprehend the repeatably regenerative non-adhesive property.

R: Thank you for your comment. We measured the sliding angles for all of them, and they were consistently higher than 20° for all surfaces.

Comments on the Quality of English Language:  Minor editing of English language required

R: The English editing was conducted on revised manuscript  by Maxafeau Professional Services - Editing.

Round 2

Reviewer 2 Report

Comments and Suggestions for Authors

I would like to thank the authors for their revisions.

Reviewer 3 Report

Comments and Suggestions for Authors

All the comments have been effectively addressed, enhancing the quality of the paper and making it suitable for publication.